# Revisiting Virtual Nodes in Graph Neural Networks for Link Prediction

## Abstract

It is well known that the graph classification performance of graph neural networks often improves by adding an artificial virtual node to the graphs, which is connected to all nodes in the graph. Intuitively, the virtual node provides a shortcut for message passing between nodes along the graph edges. Surprisingly, the impact of virtual nodes with other problems is still an open research question.

In this paper, we adapt the concept of virtual nodes to the link prediction scenario, where we usually have much larger, often dense, and more heterogeneous graphs. In particular, we use multiple virtual nodes per graph and graph-based clustering to determine the connections to the graph nodes. We also investigate alternative clustering approaches (e.g., random or more advanced) and compare to the original model with a single virtual node. We conducted extensive experiments over different datasets of the Open Graph Benchmark (OGB) and analyze the results in detail. We show that our virtual node extensions yield rather stable performance increases and allow standard graph neural networks to compete with complex state-of-the-art models, as well as with the models leading the OGB leaderboards.

## 1 Introduction

*Link prediction* is an important task to complete graphs that are missing edges in various domains: citation networks (Kipf & Welling, 2016), social networks (Adamic & Adar, 2003), medical drug interaction graphs (Abbas et al., 2021), or knowledge graphs (KGs) (Ji et al., 2021). Numerous kinds of models have been proposed to solve the link prediction problem, ranging from KG-specific predictors (Ji et al., 2021) to graph neural networks (GNNs) (Kipf & Welling, 2016; Zhang & Chen, 2018). Over dense biomedical networks, GNNs turned out to work especially well (Hu et al., 2020).

In this work, we focus on *graph neural networks* for link prediction. Many of the popular GNNs are based on the message-passing scheme, which computes node embeddings based on iteratively aggregating the features of (usually direct/one-hop) neighbor nodes along the graph edges (Gilmer et al., 2017). Interestingly, best performance is usually obtained by only considering two to three hops of neighbors (i.e., 2-3 layers in the GNN). One main reason identified for this is *over-smoothing*, the problem that node representations become indistinguishable when the number of layers increases (Li et al., 2018). The exponentially-growing amount of information has also been suggested as one issue connected to capturing long-range dependencies (Alon & Yahav, 2021). While it is likely that link prediction most often depends on the local node neighborhood, it is not beyond imagination that there are critical long-range dependencies (e.g., complex chains of drug-drug or drug-protein interactions). Hence, using a small number of layers to overcome the above problems results in *under-reaching*.

There have been several recent proposals to overcome under-reaching. On the one hand, several works propose techniques that allow for larger numbers of GNN layers (Xu et al., 2018; Wu et al., 2019; Liu et al., 2020; Chen et al., 2020; Sun et al., 2021; Zhou et al., 2020; Li et al., 2020a). However, although (Chen et al., 2020) show that over-smoothing happens particularly in dense graphs, the link prediction experiments in these works consider citation or recommendation networks, but not the especially dense biomedical ones. And our experiments over the latter suggest that the reported results are not generalizable to the more challenging biomedical data. On the other hand, there are approaches that adapt the message-passing scheme to consider neighbors beyond the one-hop neighborhood: based on graph diffusion (Atwood & Towsley, 2016; Klicpera et al., 2019a; Abu-El-Haija et al., 2019; Xu et al., 2019a; Ma et al., 2020; Klicpera et al., 2019b) and other theories (Morris et al., 2019; You

et al., 2019). However, most of these models are relatively complex and, in fact, in our experiments over the challenging graphs from the Open Graph Benchmark (OGB) (Hu et al., 2020), several ran out of memory. Moreover, the majority has not considered link prediction, while this problem was recently shown to be more difficult than node classification (Zhang et al., 2020).

In this paper, we propose a simple but elegant solution to under-reaching based on the concept of *virtual nodes* (Gilmer et al., 2017; Li et al., 2017; Pham et al., 2017; Ishiguro et al., 2019). Virtual nodes are well known to often improve the graph classification performance of graph neural networks, where an artificial virtual node is added to every graph and connected to all nodes in the graph. While the virtual nodes were originally thought as representations of the entire graph, they also provide shortcuts for message passing between nodes along the graph edges. Surprisingly, the impact of virtual nodes for the link prediction problem has not been investigated yet. The reason for this might be that the often very large and heterogeneous "network" graphs in link prediction are of very different nature and require novel/adapted solutions (e.g., a protein interaction network may easily contain millions of nodes, whereas a molecule to be classified contains usually less than fifty).

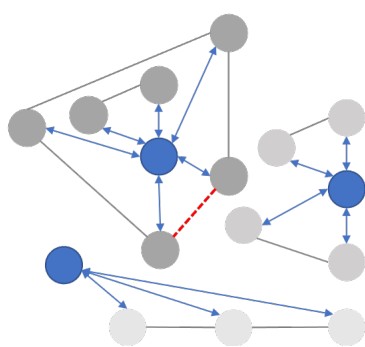

Figure 1: We cluster graph nodes belonging together (gray shades) and connect them to a common virtual node (blue). This eases information exchange and hence prediction of links between nodes not directly related (red).

We explore application and effects of virtual nodes in link prediction theoretically and empirically:

- *We propose to use multiple virtual nodes in the link prediction scenario and describe a graph-based technique to connect them to the graph nodes.* Consider Figure 1. In a nutshell, we use a graph clustering algorithm to determine groups of nodes in the graph that belong together and then connect these nodes to a common virtual node. In this way, under-reaching is decreased because clustered nodes can share information easily; at the same time, the nodes are spared of unnecessary information from unrelated nodes (i.e., in contrast to the single virtual node model).
- We also investigate alternative methods to determine the virtual node connections (e.g., randomization in clustering) and compare to the original model with a single virtual node.
- We *theoretically investigate the benefit of using (multiple) virtual nodes* in terms of two aspects: influence score and the expressiveness in learning a structural link representation.
- We conducted extensive experiments over challenging datasets of different type, provide ablation studies that confirm the superiority of our proposed techniques, analyze the results in detail, and *provide first guidelines about how to use virtual nodes with different types of data and GNNs*.
- Most importantly, we show that *our virtual node extensions most often yield rather stable performance increases* and allow standard GNNs to compete with complex state-of-the-art models that also try to improve message passing, as well as with the models leading the OGB leaderboards.

## 2  RELATED WORK

We give an overview on approaches that are similar from a technical perspective; for a more detailed summary, see Appendix A. For a more general overview of the large and diverse field of link prediction, we refer to good summaries in recent works (Martínez et al., 2016; Zhang et al., 2020).

**Deeper GNNs.**  Several techniques address over-smoothing and hence allow for constructing deeper GNNs to solve under-reaching. These models range from the simple but efficient message propagation in SGC (Wu et al., 2019; Liu et al., 2020) and APPNP (Klicpera et al., 2019a) and connections in JKNet (Xu et al., 2018), to more advanced proposals (Chen et al., 2020; Sun et al., 2021; Zhou et al., 2020; Li et al., 2020a) such as the differentiable aggregation functions in DeeperGCN (Li et al., 2020a). However, although (Chen et al., 2020) show that over-smoothing happens particularly in dense graphs, the experiments in most of these works consider citation or recommendation networks, but not the especially dense and important biomedical ones. And our experiments over the latter suggest that the reported results are not generalizable to the more challenging biomedical data.

**Beyond One-Hop Neighbors.** Recently, graph diffusion methods are used in various ways to determine the message targets and thus extend standard message passing beyond the one-hop neighborhood. Atwood & Towsley (2016) use k-hop random walks to extend the node features. APPNP (Klicpera et al., 2019a) applies personalized PageRank to propagate the node predictions generated by a neural network. Other models concatenate (Abu-El-Haija et al., 2019) or aggregate (Xu et al., 2019a; Ma et al., 2020) node embeddings in every layer using a diffusion-based transition matrix. The diffusion-based graph neural network (GDC) (Klicpera et al., 2019b) aggregates information from multiple neighborhood hops at each layer by sparsifying a generalized form of graph diffusion. Subsequent works use diffusion methods on multiple scales (Liao et al., 2019; Luan et al., 2019; Xhonneux et al., 2020) and attention (Wang et al., 2020). Morris et al. (2019) take higher-order graph structures at multiple scales into account during message passing based on the $k$-dimensional Weisfeiler and Leman graph algorithm. All the above approaches are relatively complex, many terminated with memory errors in our experiments, and few have been evaluated for link prediction.

**Virtual Nodes.** To the best of our knowledge, virtual nodes have only been considered in the context of graph classification so far, where a single virtual node (also called *supernode*) is added to the graph to be classified and connected to all graph nodes (Gilmer et al., 2017; Li et al., 2017; Pham et al., 2017; Ishiguro et al., 2019). Note that the original idea was to compute a graph embedding in parallel with the node embeddings and even connected the virtual node only in one direction (i.e, via edges from the graph nodes) instead of bidirectionally (Li et al., 2017).

There are some GNNs which point out special nodes that we could consider as "virtual". Fey et al. (2020) propose a GNN for molecule graph classification which clusters certain nodes within a molecule using a structure-based, molecule-specific algorithm and then applies message passing within and between these clusters. The graph-partition based message passing from Liao et al. (2018) also used clustering, but it just divides the original messages into inter- and intra-cluster. Our approach creates new "paths" in the graph and we theoretically demonstrate its expressiveness. P-GNN (You et al., 2019) assigns nodes to random clusters ("anchor-sets") and then creates a message for each node for every anchor-set, while ignoring the message passing from original direct neighbors. Our virtual nodes represent an alternative means to aggregate messages from multiple graph nodes which are not necessarily direct neighbors. We also explore the idea of similar random assignments in our context, but show that more elaborate techniques generally work better. Most importantly, we do not propose a specific, new GNN but a *new technique for augmenting existing graph neural networks*.

Although it is a well-known trick, the advantage of using virtual nodes has never been theoretically investigated nor fully understood. We focus on link prediction and considerably extend the virtual node technique. There are commonalities in the advantages of using virtual nodes for graph classification and link prediction, but their role in link prediction is to improve the representation of the link instead of the graph (nodes). We analyze theoretically and empirically how they improve GNN performance.

## 3 PRELIMINARIES

**Link Prediction.** We consider an undirected *graph* $G = (V, E)$ with *nodes* $V$ and *edges* $E \subseteq V \times V$. Note that this basic choice is only for ease of presentation. All our techniques work for directed graphs and, with simple adaptation, also for graphs with labelled edges. We assume $V$ to be ordered and may refer to a node by its index in $V$. For a node $v \in V$, $\mathcal{N}_v$ denotes the set of its neighbors. Given two nodes, the *link prediction task* is to predict whether there is a link between them.

**Message-Passing Graph Neural Networks.** In this paper, we usually use the term *graph neural networks* (GNNs) to denote GNNs that use message passing as described by Gilmer et al. (2017). These networks compute for every $v \in V$ a node representation $h_v^\ell$ at layer $\ell \in [1, 2, \ldots, k]$, by *aggregating* its neighbor nodes based on a generic aggregation function and then *combine* the obtained vector with $h_v^{\ell-1}$ as below; $h_v^0$ are the initial node features.

$$h_v^\ell = \text{COMBINE}^\ell \left( h_v^{\ell-1}, \text{AGGREGATE}^\ell \left( \{ h_u^{\ell-1} \mid u \in \mathcal{N}_v \} \right) \right) \tag{1}$$

Link prediction with GNNs is usually done by combining (e.g., concatenating) the final representations $h_u^L, h_v^L$, of the nodes $u, v$ under consideration and passing them through several feed-forward layers with a final sigmoid function for scoring. We follow this approach.

We further use $[1, n]$ to denote an interval $[1, 2, \ldots, n]$.

## 4 VIRTUAL NODES IN GRAPH NEURAL NETWORKS FOR LINK PREDICTION

So far, virtual nodes have been only used for graph classification. Link prediction scenarios are different in that the graphs are usually very large, heterogeneous, sometimes dense, and the task is to predict a relationship that might strongly be influenced depending on surrounding relations. In the following, we propose approaches that fit these scenarios.

### 4.1 MULTIPLE VIRTUAL NODES

Our main goal of using virtual nodes is to provide a shortcut for sharing information between the graph nodes. However, the amount of information in a graph with possibly millions of nodes is enormous, and likely too much to be captured in a single virtual node embedding. Further, not all information is equally relevant to all nodes. Therefore we suggest to use *multiple virtual nodes* $S = \{s_1, s_2 \ldots, s_n\}$[1] each being connected to a subset of graph nodes, as determined by an assignment $\sigma : V \to [1, n]$; $n$ is considered as hyperparameter. We propose different methods to obtain $\sigma$:

**Random (GNN-RM).** Most simple, we can determine a fixed $\sigma$ randomly once with initialization.

**Increased Randomness (GNN-RM$^F$).** Similarly a random assignment, but initialized with every forward pass. In this way, a single bad assignment does not determine the overall performance.

**Clustering (GNN-CM).** Many types of graph data incorporate a certain cluster structure (e.g., collaboration or social networks) that reflects which nodes belong closely together. We propose to connect such nodes in a cluster to a common virtual node, such that the structure inherent to the given graph is reflected in our virtual node assignment $\sigma$. More precisely, during initialization, we use a generic clustering algorithm which, given a number $m$, creates a set $C = \{C_1, C_2 \ldots, C_m\}$ of clusters (i.e., sets of graph nodes) by computing an assignment $\rho : V \to [1, m]$, assigning each graph node to a cluster. We then obtain $\sigma$ by choosing $m = n$ and $\sigma = \rho$.

In this work, we decided for the METIS clustering (Karypis & Kumar, 1998) which turned out to provide a good trade off between quality and efficiency. Nevertheless, our idea is generic and can be applied with arbitrary algorithms. We will show ablation experiments for alternatives (e.g., Graclus (Dhillon et al., 2007) and Diffpool (Ying et al., 2018b)).

**Advanced Clustering (GNN-CM$^+$).** Not every type of graph data contains an inherent cluster structure or one that is sufficiently expressed. Furthermore, using a fixed clustering, we obtain a deterministic algorithm again taking the risk that we completely rely on a single, possibly not ideal, virtual node assignment – there may be critical long range dependencies that go beyond clusters. For these cases, we propose an alternative approach, which breaks up the determinism by extending the above clustering as follows. We choose a relatively large $m$, with $m \gg n$, and apply the above clustering algorithm, during initialization. Then, in each epoch, we randomly guess an assignment $\sigma' : [1, m] \to [1, n]$ of clusters to virtual nodes and define $\sigma(n) := \sigma'(\rho(n))$. Note that this approach is inspired by Chiang et al. (2019), who apply a similar technique is to create batches based on clusters. Further note that we determine $\sigma'$ with every epoch instead of every forward pass since the the computation takes quite some time on large datasets and we observed that this yields good results.

### 4.2 THE MODEL

We integrate the multiple virtual nodes into a generic message-passing graph neural network by extending the approach from Hu et al. (2020) to the setting with multiple virtual nodes, by computing node representations $h_v^\ell$ for a node $v \in V$ at layer $\ell$ as follows:

$$h_{s_i}^\ell = \text{COMBINE}_{VN}^\ell \left( h_{s_i}^{\ell-1}, \text{AGGREGATE}_{VN}^\ell \left( \{ h_u^{\ell-1} \mid u \in V, \sigma(u) = i \} \right) \right) \tag{2}$$

$$h_v^\ell = \text{COMBINE}^\ell \left( h_v^{\ell-1} + h_{s_{\sigma(v)}}^\ell, \text{AGGREGATE}^\ell \left( \{ h_u^{\ell-1} \mid u \in \mathcal{N}_v \} \right) \right) \tag{3}$$

Note that the highlighted adaptation of the standard GNN processing from Equation (1) is only minor – but powerful. In our implementation, $\text{COMBINE}_{VN}^\ell$ is addition combined with linear layers and layer normalization, and we use sum for $\text{AGGREGATE}_{VN}^\ell$.

---

[1]Since notation $V$ is standard for nodes, we use $S$ for the set of virtual nodes. Think of "supernodes".

### 4.3 Analysis: Virtual Nodes Change Influence

**Influence Score.** Following (Xu et al., 2018; Klicpera et al., 2019a), we measure the sensitivity (also, *influence*) of a node $x$ to a node $y$ by the *influence score* $I(x, y) = e^{\mathrm{T}} \frac{\partial h_x^k}{\partial h_x^0}$; $e$ is a vector of all ones, $h_x^k$ is the embedding of $x$ at the $k^{th}$ layer, see Equations (1) and (3). For a $k$-layer GNN, the influence score is known to be proportional in expectation to the $k$-step random walk distribution from $x$ to $y$:[2]

$$\mathbb{E}[I(x, y)] \propto P_{rw}(x \to y, k) = \sum_{r \in R^k} \prod_{\ell=1}^{k} \frac{1}{deg(v_r^\ell)}, \tag{4}$$

$(v_r^0, v_r^1, ..., v_r^k)$ are the nodes in the path $r$ from $x := v_r^0$ to $y := v_r^k$, $R^k$ is the set of paths of length $k$. In what follows, we will exploit this relationship and argument in terms of the probability $P_{rw}$.

**Virtual Nodes.** For simplicity, consider the influence score in an $m$-regular graph; there we have $P_{rw}(x \to y, k) = \frac{|R^k|}{m^k}$. We hypothesize that we can come to similar conclusions in a general graph with average degree $m$. Consider the message passing between two distant nodes $x$ and $y$. (I) In case the shortest path from $x$ to $y$ is of length $> k$, a $k$-layer GNN cannot capture it, and the probability $P_{rw}(x \to y, k)$ is obviously zero. If we then consider virtual nodes in the GNN layer (even with only one), we can pass messages from $x$ to $y$ through the virtual nodes and obtain a nonzero probability. (II) Consider the case where there is a shortest path of length $\leq k$ between $x$ and $y$. By adding a virtual node $s$ in one GNN layer, the probability changes to:

$$P_{rw}^s(x \to y, k) = P_{rw}(x \to y, k) + P_{rw}(x \to s, s \to y) = \frac{|R^k|}{(m+1)^k} + \frac{1}{(m+1)|V|}. \tag{5}$$

Compared to the original probability, we get the following impact ratio for using virtual nodes:

$$ir = \frac{m^k}{(m+1)^k} + \frac{m^k}{(m+1)|V||R^k|}. \tag{6}$$

When $m$ is large enough, $ir$ can be approximated by $ir \simeq \left(1 + \frac{m^{k-1}}{|V||R^k|}\right)$. Here, we see that the impact of virtual nodes grows when $m$ increases. Our experiments confirm this theoretical observation.

**Multiple Virtual Nodes.** In view of multiple virtual nodes, the above analysis gets even more appealing. We continue along these lines and assume there is a shortest path of length $\leq k$ between $x$ and $y$. If $x$ and $y$ connect to the same virtual node $s$, then Equation (5) changes as follows:

$$P_{rw}^s(x \to y, k) = \frac{|R^k|}{(m+1)^k} + \frac{1}{(m+1)|C_s|}. \tag{7}$$

Since the set $C_s$ of nodes connecting to $s$ is much smaller than $V$, the impact of multiple virtual nodes is greater than that of a single virtual node. On the other hand, if $x$ and $y$ do not connect to the same virtual node, the probability just slightly decreases from $\frac{|R^k|}{m^k}$ to $\frac{|R^k|}{(m+1)^k}$.

In Appendix B, we further show that using multiple virtual nodes is related to (but not equal to) the labeling trick (Zhang et al., 2020) and distance encoding (Li et al., 2020b), and it can theoretically improve the expressiveness in learning structural link representations (see Theorem 1 and Figure 4(b)).

## 5 Evaluation

We conducted extensive experiments and ablation studies to empirically investigate:

- How does the existing approach with **one virtual node perform in link prediction**?
- **Do multiple virtual nodes improve performance**, how do our proposed approaches compare?
- In particular, **are approaches based on the graph structure better**?
- How exactly do virtual nodes support link prediction? **When do they help particularly?**

---

[2]See Theorem 1 in (Xu et al., 2018). Note that the theorem makes some simplifying assumptions (e.g., on the shape of GNN).

Table 1: Data. All graphs are undirected, have no edge features, and all but `ddi` have node features.

|          | #Nodes  | #Edges     | Average Node Deg. | Average Clust. Coeff. | MaxSCC Ratio | Graph Diameter |
|----------|---------|------------|-------------------|------------------------|--------------|----------------|
| ddi      | 4,267   | 1,334,889  | 500.5             | 0.514                  | 1.000        | 5              |
| ppa10    | 509,860 | 11,217,535 | 8.3               | 0.019                  | 0.983        | 29             |
| pubmed   | 19,717  | 88,648     | 4.5               | 0.046                  | 0.992        | 20             |
| collab   | 235,868 | 1,285,465  | 8.2               | 0.729                  | 0.987        | 23             |

Table 2: GCN, SAGE, and GIN with different virtual node configurations over different graph-data types/amounts; (second) best results are (light) gray, overall best **bold**, second best underlined.

|          | ddi Hits@20           | ppa10 Hits@100        | collab Hits@50        | pubmed Hits@20        |
|----------|-----------------------|-----------------------|-----------------------|-----------------------|
| GCN      | $0.4076 \pm 0.1073$   | $0.1313 \pm 0.0084$   | $0.4955 \pm 0.0064$   | $0.9675 \pm 0.0143$   |
| - VN     | $0.6217 \pm 0.1241$   | $0.1258 \pm 0.0082$   | $0.5049 \pm 0.0088$   | $0.9579 \pm 0.0214$   |
| - RM     | $0.5532 \pm 0.1262$   | $0.1205 \pm 0.0059$   | $0.5083 \pm 0.0109$   | $0.9522 \pm 0.0110$   |
| - $RM^F$ | $0.5830 \pm 0.0855$   | $0.1116 \pm 0.0094$   | $0.5046 \pm 0.0049$   | $0.8100 \pm 0.0781$   |
| - CM     | $0.6105 \pm 0.1563$   | $0.1299 \pm 0.0050$   | $0.5181 \pm 0.0076$   | $0.9575 \pm 0.0230$   |
| - $CM^+$ | $0.6033 \pm 0.1759$   | $0.1399 \pm 0.0071$   | $0.5128 \pm 0.0129$   | $0.9189 \pm 0.0514$   |
| SAGE     | $0.6173 \pm 0.1068$   | $0.1024 \pm 0.0050$   | $0.5662 \pm 0.0149$   | $0.9779 \pm 0.0105$   |
| - VN     | $0.6491 \pm 0.1360$   | $0.0853 \pm 0.0154$   | $0.5875 \pm 0.0091$   | $0.9659 \pm 0.0333$   |
| - RM     | $0.7068 \pm 0.1174$   | $0.1131 \pm 0.0039$   | $0.5830 \pm 0.0087$   | $0.9433 \pm 0.0208$   |
| - $RM^F$ | $0.7564 \pm 0.1055$   | $0.1105 \pm 0.0023$   | $\mathbf{0.6067 \pm 0.0063}$ | $0.9800 \pm 0.0087$ |
| - CM     | $0.7621 \pm 0.1157$   | $0.1077 \pm 0.0150$   | $0.6056 \pm 0.0105$   | $\mathbf{0.9834 \pm 0.0068}$ |
| - $CM^+$ | $\mathbf{0.8251 \pm 0.0678}$ | $0.0963 \pm 0.0099$ | $0.5940 \pm 0.0262$ | $0.9754 \pm 0.0139$ |
| GIN      | $0.4321 \pm 0.1353$   | $0.1139 \pm 0.0058$   | $0.5768 \pm 0.0179$   | $0.9234 \pm 0.0166$   |
| - VN     | $0.5260 \pm 0.1227$   | $0.1316 \pm 0.0049$   | $0.5863 \pm 0.0254$   | $0.9790 \pm 0.0070$   |
| - RM     | $0.5084 \pm 0.1324$   | $0.1337 \pm 0.0045$   | $0.5412 \pm 0.0174$   | $0.9604 \pm 0.0158$   |
| - $RM^F$ | $0.5310 \pm 0.1453$   | $0.1269 \pm 0.0026$   | $0.5335 \pm 0.0087$   | $0.7986 \pm 0.0993$   |
| - CM     | $0.5664 \pm 0.0860$   | $0.1349 \pm 0.0034$   | $0.5821 \pm 0.0081$   | $0.9125 \pm 0.0378$   |
| - $CM^+$ | $0.4339 \pm 0.1855$   | $\mathbf{0.1591 \pm 0.0069}$ | $0.5557 \pm 0.0026$ | $0.9037 \pm 0.0262$ |

**Datasets.** We focused on challenging data from the OGB: `ddi`, a drug-drug interaction network; `ppa10`, a subset of the protein-protein association network `ppa` containing only 10% of the train edges (but full valid/test); and `collab`, an author collaboration network. To learn more about smaller data of similar type, we also tested on the citation networks `pubmed` (Yang et al., 2016). Since the datasets are not only very different in type but also in various other critical graph parameters and this is reflected in the performance of the models, we show relevant statistics in Table 1.[3] The datasets vary strongly in size with `ddi` being smallest among the biomedical; on the other hand, `ddi` is very dense. The clustering coefficient intuitively reflects the "cliquishness" of the graph's subgraphs. The large diameters suggest that the data suits testing under-reaching. Appendix C gives further details and describes datsets we consider in additional experiments in the appendix.

**Baselines** For a competitive comparison, we considered important baselines (described in Section 2):

- The deep GNNs **SGC**, **APPNP**, **DeeperGCN**, and two variants of **JKNet**.
- Approaches extending message passing: **P-GNN**, **APPNP**, **GCN-GDC**, **SAGE-GDC**, **GIN-GDC**.
- The popular GNNs **GCN** (Kipf & Welling, 2017), **SAGE** (Hamilton et al., 2017), and **GIN** (Xu et al., 2019b), which we then extend with (multiple) virtual nodes.

---

[3]See Tables 2 and 3 in (Hu et al., 2020). We computed the numbers for `ppa10` (which we focus on due to a lack of resources), and `pubmed` using the same techniques.

Table 3: Comparison to SOTA; **top**: state of the art (two best per dataset) according to OGB leaderboard (10/3/21), leaderboard results are marked by *; **middle**: models with similar goal to our approach ("-": ran out of memory); **bottom**: best of virtual-node augmented GNNs we tested.

| | ddi 
 Hits@20 | collab 
 Hits@50 |
|---|---|---|
| SAGE (adapted)+attr.* | $\mathbf{0.8781 \pm 0.0474}$ | n/a |
| SAGE +dist.* | $0.8239 \pm 0.0437$ | n/a |
| Adamic Adar+* | $0.1861 \pm 0.0000$ | $\mathbf{0.6548 \pm 0.0000}$ |
| SEAL* | $0.3056 \pm 0.0386$ | $\underline{0.6474 \pm 0.0043}$ |
| DeeperGCN* | n/a | $0.6187 \pm 0.0045$ |
| GCN+JKNet* | $0.6056 \pm 0.0869$ | n/a |
| SGC | $0.0676 \pm 0.0586$ | $0.4635 \pm 0.0197$ |
| P-GNN | $0.1050 \pm 0.0000$ | - |
| APPNP | $0.1492 \pm 0.0298$ | $0.3185 \pm 0.0205$ |
| GCN-GDC | $0.2550 \pm 0.1242$ | - |
| SAGE-GDC | $0.3141 \pm 0.1254$ | - |
| GIN-GDC | $0.2180 \pm 0.0786$ | - |
| SAGE | $0.6173 \pm 0.1068$ | $0.5516 \pm 0.0171$ |
| SAGE-CM$^+$/CM | $\underline{0.8251 \pm 0.0678}$ | $0.6017 \pm 0.0137$ |

## 5.1 RESULTS

**Overall Impact of Virtual Nodes, Tables 2, 7, 8 (Appendix).** We compare to GCN, SAGE, and GIN. The common approach of using a single virtual node (GNN-VN) yields good improvements over `ddi`, slight improvements over `collab`, but no definitive ones over `ppa10`; over `pubmed`, it works very well for GIN. The numbers for GNN-RM and GNN-RM$^F$ reflect the randomness of their connections to the virtual nodes, there is no clear trend. Nevertheless, they clearly outperform the original models, with only few exceptions. The increased randomness by re-assigning the virtual nodes with every forward pass (GNN-RM$^F$) seemingly suits SAGE but not the others. As expected, over the small `pubmed`/`cora`, which also have no cluster structure, the results are not consistent or convincing overall; virtual nodes only yield improvement sometimes, and none for GCN. Yet, on the more challenging datasets, *multiple virtual nodes turn out to be an efficient means to boost the link prediction performance of GNNs* if they are applied correctly. Our virtual node connections based on the graph structure (GNN-CM) yield consistently good improvements over `ddi` and `collab`, and mostly help on the challenging `ppa10` dataset. On `collab`, we did further experiments using GAT (Veličković et al., 2017) and also observe a clear performance gain: 0.4745 vs. 0.5876 (GAT-CM). GNN-CM and GNN-CM$^+$ are not always the best ones, but yield reliably good results, in contrast to the other models with virtual nodes (see variability of gray shades). Interestingly, the advanced clustering yields especially good performance over `ppa10`/`ppa`, while its results on the other datasets are not convincing. Generally, the improvements of the virtual node models are strongest on `ddi`. For an in-depth result analysis see Section 5.2.

**Comparison to Related Works and SOTA, Table 3.** Most deep GNNs as well as the models that use complex message-passing techniques perform disappointing and, overall, much worse than the standard GNNs. We did thorough hyperparameter tuning for these models and it is hard to explain. However, most of the original evaluations focus on node or graph classification and consider very different types of data – often the standard citation networks (Lu & Getoor, 2003) – and, in fact, on `collab` we see the best numbers. For a more detailed discussion of P-GNN see Appendix H. Even if we assume that these numbers can be improved, the models do not seem apt for link prediction; in particular, the complex ones: many do not run at all on realistic link prediction data but yield memory errors. Further, *our virtual node extensions make standard GNNs competitive to the models on the leaderboard.* In particular, their performance is much more stable. The results of the best models from the leaderboard vary strongly with the different datasets, or have not been reported at all. None of these models can be called "good" overall, given the numbers in the - sometimes even missing - rest of the table; in fact, SEAL and Adamic Adar perform rather bad on the very dense `ddi`.

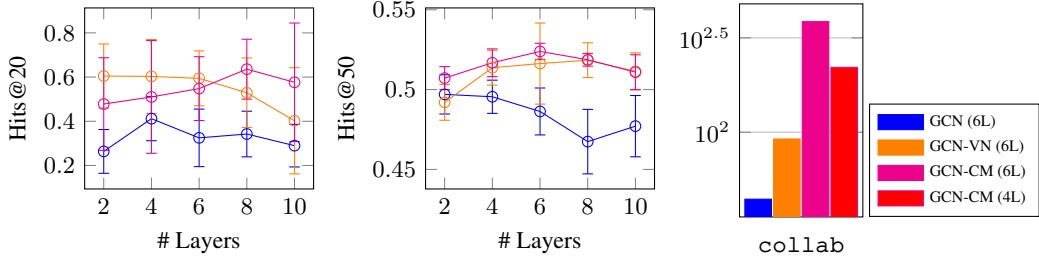

Figure 2: Performance depending on layers: Hits@k and time per epoch (sec.); `ddi` (left), `collab`.

**Impact of Virtual Nodes on Number of GNN Layers and Efficiency, Figure 2.** For the virtual nodes models, the scores increase with the number of layers for a longer time, GCN drops earlier. On `ddi`, GCN-VN and -CM reach their best scores at 6 and 8 layers, respectively, which is remarkable for that very dense dataset, which is prone to over-smoothing. On `collab` it is the other way around. The figure also gives an idea about the runtime increase with using virtual nodes. It compares the 6-layer models, and shows the 4-layer GCN-CM which obtains performance similar to the 6-layer GCN-VN.

**Impact of Virtual Node Number, Figure 3.** First, consider the configurations of the best models for the overall results in Table 2, which are provided in Table 6 in the appendix. Here, we see that the chosen numbers of virtual nodes are indeed random for the "random" models, but GNN-CM consistently uses a high number of virtual nodes, which also suits it better according to our theoretical analysis in Section 4.3. In line with this, the more detailed analysis varying the numbers of virtual nodes, yields best results (also in terms of standard deviations) for SAGE-CM at rather high values. For GCN, we do not see a clear trend, but (second) best performance with 64 virtual nodes. Note that there is a trade off between number of virtual nodes and intra-cluster test edges, discussed in Section 5.2.

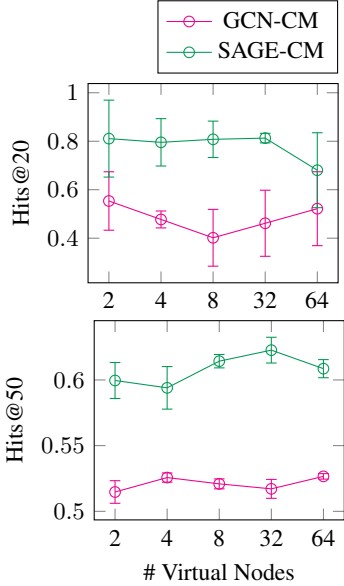

Figure 3: Impact of virtual node number; `ddi` (top) and `collab`.

**Using Virtual Nodes Only at the Last GNN Layer, Table 4.** Alon & Yahav (2021) show that using a fully connected adjacency matrix at the last layer of a standard GNN helps to better capture information over long ranges. We therefore investigated if it is a better architectural choice to use virtual nodes only at the last layer. However, we see that this can lead to extreme performance drops.

**Impact of Clustering Algorithm, Table 9 (Appendix).** Our architecture is generic in the clustering algorithm, and we investigated the effects of varying that. Graclus is similar in nature to METIS in that it also creates partitions based on the adjacency matrix, but it took much longer to run. Diffpool considers the node features and yields improvements for GCN, but does not scale to larger datasets. Over `ddi`, there is no clear winner and, given its efficiency, METIS turns out to be a good solution.

Table 4: Comparison of using the virtual nodes at every and only at the last layer; Hits@20, `ddi`.

|  | GCN | SAGE | GIN |
|---|---|---|---|
| w/o virtual nodes | $0.5062 \pm 0.2186$ | $0.6128 \pm 0.2122$ | $0.4829 \pm 0.1608$ |
| - VN | $0.5932 \pm 0.2390$ | $\underline{0.7160 \pm 0.1457}$ | $0.6523 \pm 0.0446$ |
| - VN$_{OL}$ | $0.6180 \pm 0.0088$ | $0.5167 \pm 0.1364$ | $\underline{0.6472 \pm 0.0542}$ |
| - CM | $\underline{0.6322 \pm 0.1565}$ | $\mathbf{0.8819 \pm 0.0341}$ | $\mathbf{0.6544 \pm 0.0960}$ |
| - CM$_{OL}$ | $\mathbf{0.6338 \pm 0.1188}$ | $0.6151 \pm 0.1545$ | $0.4420 \pm 0.1694$ |

## 5.2 DISCUSSION AND CONCLUSIONS

The results show that our approach with multiple virtual nodes based on graph-based clustering yields performance increases for various GNNs and types of data, but there are clear differences.

**Dense Graphs with Medium/High Clustering Coefficient.** Over `ddi`, we see strongest improvements for all virtual-node models. This can be explained by our proposed theory, showing that a very large node degree $m$ increases the impact of the virtual node(s), and thus decreases the negative impact of the (too) many other neighbors (see Equation (6)). Furthermore, the empirical results confirm our proposed theory regarding multiple virtual nodes (see Equation (7)). We see particularly good numbers for GNN-CM, which exploits the clustering inherent in the given graph. GNN-CM$^+$, which considers this given clustering only on a lower level, is shown to perform worse than GNN-CM overall. In fact, we computed the percentage of test edges that occur in the "virtual node cluster" (see Table 11 in the appendix) and it shows that the numbers for the advanced clustering are very similar to the random one, meaning the randomly merged smaller clusters break the data's structure too much. Interestingly, the experiments show that, even with the dense data that is prone to over-smoothing, virtual nodes make the GNNs score higher with more than the standard 2-3 layers; hence virtual nodes seem to alleviate over-smoothing to some extent, an interesting question for future work.

**Graphs with Large Problem Radius and Low Clustering Coefficient.** Over `ppa10`, all GNNs use an unusually high number of layers, which hints at a large problem radius (e.g., GCN, which performs especially well, uses 7 layers). Given the very low clustering of the data in addition, `ppa10` represents a special challenge. With the multiple virtual nodes, GNN-CM performs again better than GNN-VN. On the other hand, it does not perform much better than the random models on data without cluster structure. This can be explained by its choice of number of virtual nodes, which is consistently high, but then there are less test edges within a virtual node cluster (see appendix Table 11). We hence see here that the positive effect of having many virtual nodes (recall Equation (7)) cancel out the benefits of clustering. Our advanced clustering, which merges some local clustering with randomness, is able to achieve best results with GCN and GIN (with SAGE, all models perform rather bad over `ppa10`). This can be explained by the fact that it randomly merges some local clusters – with each epoch anew – and hence allows more messages to pass across "virtual node clusters". We also did some experiments over the very large `ppa`, which is denser than `ppa10`, and see a similar trend.

**Sparse Graphs with Low to High Clustering Coefficient.** We tested on three citation/collaboration networks of different sizes. Note that, over this data, the problem radius is usually assumed to be rather small (Alon & Yahav, 2021), although the graph diameters are large. We investigated virtual nodes to augment link prediction in large and complex graphs; but we also want to provide insight into the behavior on smaller data. Over `pubmed` (similarly on `cora` as shown in the appendix), virtual nodes do not provide any improvement for GCN. For GIN, a single virtual node yields good increases; overall, it usually outperforms the settings with multiple virtual nodes. We hypothesize that this is mainly due to the small graph size and sparsity. In fact, on the larger and denser `collab`, GNN-CM performs very good for all GNNs. The trends in the models' performance and the corresponding explanations are similar to those for `ddi` but much less expressed, probably due to the much smaller node degrees. Yet, the performance is much more stable, possibly because it is larger and not as dense.

**Conclusions.** We summarize our main findings to provide first guidelines for applying virtual nodes:

- **Small + Sparse Graphs:** A single virtual node is likely to boost performance of GIN, and virtual nodes should help with SAGE, but probably not with GCN.
- **Large + Sparse Graphs**: If there is cluster structure, GNN-CM should yield stable performance increases. If the problem radius is large or there is few cluster structure, GNN-CM$^+$ is worth a try.
- **Dense Graphs + Clustering:** Multiple virtual nodes (i.e., GNN-CM) likely increase performance.

## 6 CONCLUSIONS

We propose a simple but elegant graph neural network extension using multiple virtual nodes that may considerably increase link prediction performance. We also advance research by providing theoretical justifications - the very first about applying virtual nodes at all - and by showing their positive impact in various experiments. Future work includes the design of more advanced and scalable architectures, and it would be interesting to further investigate the huge performance increases on dense graphs.

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

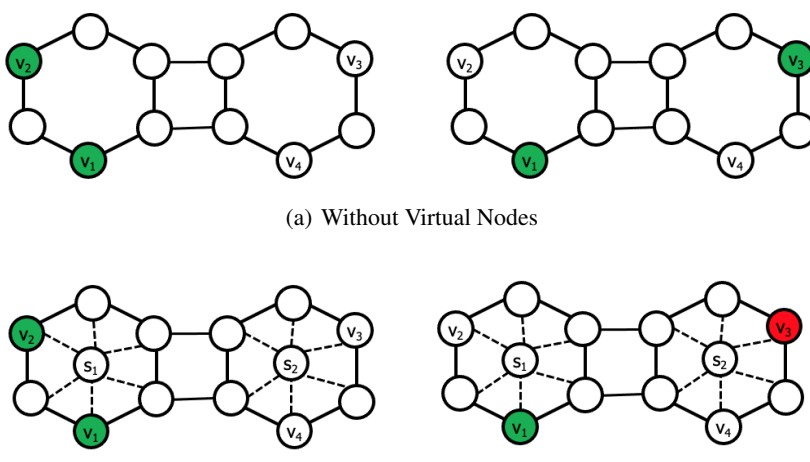

(a) Without Virtual Nodes

(b) With 2 Virtual Nodes

Figure 4: Example of using two virtual nodes to improve expressiveness. In Figure 4(a), a typical message-passing GNN (see Section 3) computes the same representation for $v_2$ and $v_3$, and we cannot discriminate the node pairs $(v_1, v_2)$ and $(v_1, v_3)$ if we just use the node representation for link prediction, as it is usual. In Figure 4(b), by adding two virtual nodes $s_1$ and $s_2$, $v_2$ and $v_3$ have clearly different features (because they connect to different virtual nodes with different labels) and different representations after GNN embedding, so $(v_1, v_2)$ and $(v_1, v_3)$ can be easily discriminated.

## A    ADDITIONAL DETAILS ON RELATED WORKS

**Deeper GNNs.**  We mention simpler approaches in the Section 2. More advanced proposals are, for example, based on special features and connections (Chen et al., 2020), community-based normalization of node representations using random clustering (Zhou et al., 2020), boosting techniques (Sun et al., 2021), or differentiable aggregation functions in DeeperGCN (Li et al., 2020a).

**Beyond One-Hop Neighbors.**    Graph diffusion methods (i.e., in graph theory, techniques for spreading information between nodes) are used in various ways to determine the message targets and thus extend standard message passing beyond the *one-hop* neighborhood. (Atwood & Towsley, 2016) use k-hop random walks to aggregate node features and extend the latter by the aggregated ones. APPNP (Klicpera et al., 2019a) applies personalized PageRank to propagate the node predictions generated by a neural network. Other models aggregate node embeddings in every layer, GraphHeat (Xu et al., 2019a) using the heat kernel, PAN (Ma et al., 2020) the transition matrix of maximal entropy random walks, and PinSage (Ying et al., 2018a) using random walks. (Abu-El-Haija et al., 2019) propose to concatenate embeddings aggregated using the transition matrices of k-hop random walks before applying one-hop neighbor aggregation. The diffusion-based graph neural network (GDC) (Klicpera et al., 2019b) aggregates information from multiple neighborhood hops at each layer by sparsifying a generalized form of graph diffusion. Subsequent works use diffusion methods on multiple scales (Liao et al., 2019; Luan et al., 2019; Xhonneux et al., 2020). Recently, (Wang et al., 2020) integrated attention with diffusion-based message propagation.

**Position Encodings.** Our approach provides a kind of positional embedding (Srinivasan & Ribeiro, 2019) and hence has some commonalities with models extending nodes with positional encodings, e.g., (Li et al.).

## B    ADDITIONAL THEORETICAL RESULTS: STRUCTURAL LINK REPRESENTATION

Adding structure-related features such as a distance encoding (Li et al., 2020b) has been demonstrated to make graph representation learning more powerful. For link prediction, (Zhang et al., 2020) propose the *labeling trick* extending distance encoding and making GNNs learn better link representations.

We first recall the definitions from (Zhang et al., 2020) introducing the concept of labeling trick. Consider an undirected graph $G$ as described in Section 3. In addition, the tensor $\mathbf{A} \in \mathbb{R}^{n \times n \times k}$ contains all node and edge features (if available). The diagonal components $\mathbf{A}_{v,v,:}$ denote the node features, while the off-diagonal components $\mathbf{A}_{u,v,:}$ denote the edge features of edge $(u, v)$. The labeling trick uses a target node set $S \subseteq V$ and a labeling function to label all nodes in the node set $V$ and stack the labels with $\mathbf{A}$. A valid labeling trick must meet two conditions: (1) the nodes in $S$ have different labels from the rest of the nodes, (2) the labeling function must be permutation invariant.

Let us recall our method using multiple virtual nodes. Assume we have multiple virtual nodes $S = \{s_1, ..., s_m\}$. $\forall u \in V$, we have the additional features for the node $l(u|S) = (h(s_1), ..., h(s_m))^{\mathrm{T}}(\gamma(u|s_1), ..., \gamma(u|s_m))$, where $\gamma(u|s_i) = 1$ if $u$ is connected to the virtual node $s_i$, and $\gamma(u|v_i) = 0$ otherwise. $h(s_i)$ is the node representation of virtual node $s_i$, and is initialized by one-hot vectors so that each virtual node has different labels.

Our labeling strategy is not a valid labeling trick by the definition of (Zhang et al., 2020). First, $S$ is not a subset of $V$, and we use addition instead of concatenation. Even if we extend $V$ to $V \cup S$, our labeling strategy still does not fit the permutation-invariant requirement. Nevertheless, it can achieve similar effects in learning structural link representations.

**Theorem 1.** *In any non-attributed graphs with $n$ nodes, if the degree of each node in the graph is between 1 and $\mathcal{O}(\log^{\frac{1-\epsilon}{2h}}(n))$ for any constant $\epsilon > 0$, given $m$ virtual nodes which evenly divide the node set into $m$ clusters, then there exists $\omega\left((m-1)^2(\frac{n^\epsilon}{m}-1)^3\right)$ many pairs of non-isomorphic links $(u, w), (v, w)$, such that an $h$-layer 1-WL-GNN (see definitions in (Li et al., 2020b) and (Zhang et al., 2020); one well-known example is GIN (Xu et al., 2019b)) gives $u, v$ the same representation, while using $m$ virtual nodes can give $u, v$ different representations.*

*Proof.* The proof can be separated into two steps. The first step is to prove that there exists $n/o(n^{1-\epsilon}) = \omega(n^\epsilon)$ many nodes that are locally $h$-isomorphic. This step is same as the proof of Theorem 2 in (Zhang et al., 2020), so we omit the details here. After getting these locally isomorphic nodes, we denote the set of these nodes as $V_{iso}$. The second step is to find the non-isomorphic links.

**Step 2.** Let us partition $V_{iso} = \cup_{i=1} V_i$ where $V_i$ is the subset of nodes connected to virtual node $s_i$. To be simple, we call each $V_i$ a cluster, and the sizes of different clusters are assumed to be the same $|V_i| = |V_{iso}|/m$. Consider two nodes $u \in V_i$ and $v \in V_j$ from different clusters. Since both of them are in $V_{iso}$, so they have identical $h$-hop neighborhood structures, and $h$-layer 1-WL-GNN will give them the same representations. Then let us select another node $w$ in $V_i$, $h$-layer 1-WL-GNN will also make $(u, w)$ and $(v, w)$ have the same representation.

However, if we use virtual nodes to label nodes and give them additional features, because $u, w$ are in the same cluster while $v, w$ belong to different clusters, $(u, w)$ will have different representation from $(v, w)$. Now let us count the number of such non-isomorphic link pairs $Y$, we can have:

$$Y \geq \prod_{i,j=1, j\neq i}^{m} |V_i||V_i - 1||V_j| = \frac{1}{2}m(m-1)\left(\left(\frac{|V_{iso}|}{m} - 1\right)\left(\frac{|V_{iso}|}{m}\right)^2\right)$$

Taking $|V_{iso}| = \omega(n^\epsilon)$ into the above in-equation, we get

$$Y \geq \frac{1}{2}m(m-1)\omega\left((\frac{n^\epsilon}{m} - 1)^3\right) = \omega\left((m-1)^2(\frac{n^\epsilon}{m} - 1)^3\right)$$

**Example (Power of Using Multiple Virtual Nodes).** In Figure 4, we show two cases with and without virtual nodes. Consider the nodes $v_2, v_3$ with the same local structure, which means they can get the same node representations by using 1-WL-GNN. So we cannot discriminate the links $(v_1, v_2)$ and $(v_1, v_3)$ if we just use 1-WL-GNN and concatenate the node representations for link prediction. However, if we add 2 virtual nodes and add extra features to each node. $v_1$ and $v_2$ get a new feature $(1, 0)$, $v_3$ get new feature $(0, 1)$. So it is easy to see $(v_1, v_2)$ and $(v_1, v_3)$ now have different representations.

## C    ADDITIONAL DETAILS ON THE DATA

See Table 5 for the datasets we consider additionally in the appendix.

Table 5: Data from additional experiments. All graphs are undirected, and have no edge but node features.

|  | #Nodes | #Edges | Average Node Deg. | Average Clust. Coeff. | MaxSCC Ratio | Graph Diameter |
|---|---|---|---|---|---|---|
| ppa | 576,289 | 30,326,273 | 73.7 | 0.223 | 0.999 | 29 |
| cora | 2,708 | 10,556 | 3.9 | 0.192 | 0.906 | 20 |

The datasets vary strongly in the number of nodes and edges with `cora < pubmed < ddi < collab < ppa10 < ppa`. Yet, `ddi` shows a challenging average node degree and hence is very dense. The average clustering coefficient intuitively reflects the extent to which direct neighbors of a node are themselves direct neighbors, and thus measures the "cliquishness" of the graph's subgraphs. Here, we have `ppa10 < pubmed < cora < ppa < ddi < collab`. Finally, the graph diameter varies similarly, and again shows that `ppa` represents a special challenge. The diameter does not necessarily reflect the problem depth directly, but it shows which maximal problem depth can be expected. In particular, the depths here hint at much larger numbers of GNN layers than the 2-3 layers we usually use, hence the data might suit testing under-reaching.

## D    MODEL CONFIGURATIONS AND TRAINING

We trained all models for 80 runs using the Bayesian optimization provided by wandb[4] and the following hyperparameters.

| | |
|---|---|
| hidden dimension | 32, 64, 128, 256 |
| learning rate | 0.1, 0.05, 0.01, 0.005, 0.001, 0.0005, 0.0001 |
| dropout | 0, 0.3, 0.6 |
| # of layers | 1-7 |
| # of virtual nodes (random) | 1-10 |
| # of virtual nodes | 1,2,4,8,16,32,64 |
| SGC - K | 2-7 |
| APPNP - $\alpha$ | 0.05, 0.1, 0.2, 0.3 |
| GNN-GDC - k | 64, 128 |
| GNN-GDC - $\alpha$ | 0.05, 0.1, 0.2, 0.3 |

Please note that we considered the wide ranges of values only in order to find a good general setting. For practical usage a hidden dimension of 256, learning rate of 0.0001, and dropout of 0.3 should work well; only on the small graphs a dropout of 0 might work better. As usual, the number of layers depends on the type of data; however, note that the virtual nodes make it possible to use more that then the usual 2-3 layers. Generally, higher numbers of virtual nodes work better, in line with our theoretical results.

Also note that we used less virtual nodes in the selection for the models (-RM, -RM$^F$) since especially -RM$^F$ was very slow and preliminary results showed that larger numbers did not change the results greatly – probably due to the randomness. We used maximally 64 virtual nodes due to memory issues with larger numbers (e.g., 128), especially on the larger datasets. We report the specific numbers of GNN layers and virtual nodes used by the trained models from Tables 2, 8, and 3 in Table 6. For the first clustering in GNN-CM$^+$, we created 150 clusters on `cora` and `pubmed`, 200 clusters on `ddi` and `collab`, and 1000 on `ppa10`.

We tuned all models for 80 runs, and thereafter ran the models with the best 3 configurations for 3 runs and chose the best of these model as the final model (configuration).

We trained as suggested by the OGB (e.g., the splits, negative sampling) but used a batch size of $2^{12}$ and sometimes adapted the number of runs due to lack of resources; we used 3 for the experiments on `collab` and `ppa10` in Table 2. However, we ran several of our models for 10 runs as required for results on the OGB leaderboards and the numbers are comparable (see Table 10).

---

[4]https://wandb.ai/site

Table 6: Numbers of GNN layers and virtual nodes used by the trained models from Table 2.

| | ddi | | ppa10 | | collab | | pubmed | |
|---|---|---|---|---|---|---|---|---|
| | #Lay | #VNs | #Lay | #VNs | #Lay | #VNs | #Lay | #VNs |
| GCN | 2 | 0 | 7 | 0 | 2 | 0 | 2 | 0 |
| - VN | 2 | 1 | 7 | 1 | 6 | 1 | 5 | 1 |
| - RM | 1 | 7 | 7 | 2 | 7 | 7 | 4 | 3 |
| - RM$^F$ | 2 | 9 | 6 | 9 | 3 | 2 | 5 | 8 |
| - CM | 4 | 64 | 6 | 64 | 6 | 32 | 6 | 2 |
| - CM$^+$ | 3 | 64 | 5 | 32 | 7 | 8 | 5 | 4 |
| SAGE | 2 | 0 | 4 | 0 | 5 | 0 | 2 | 0 |
| - VN | 4 | 1 | 6 | 4 | 4 | 1 | 4 | 1 |
| - RM | 4 | 10 | 5 | 6 | 7 | 4 | 7 | 3 |
| - RM$^F$ | 4 | 3 | 6 | 9 | 7 | 3 | 4 | 10 |
| - CM | 3 | 8 | 6 | 64 | 5 | 32 | 4 | 16 |
| - CM$^+$ | 3 | 16 | 6 | 4 | 6 | 2 | 7 | 8 |
| GIN | 6 | 0 | 2 | 0 | 2 | 0 | 5 | 0 |
| - VN | 5 | 1 | 7 | 1 | 4 | 1 | 6 | 1 |
| - RM | 2 | 5 | 6 | 8 | 3 | 7 | 5 | 10 |
| - RM$^F$ | 1 | 4 | 4 | 7 | 3 | 9 | 5 | 7 |
| - CM | 4 | 64 | 4 | 64 | 3 | 64 | 7 | 4 |
| - CM$^+$ | 5 | 1 | 6 | 16 | 4 | 16 | 4 | 1 |

Table 7: Results on `ppa`: OGB leaderboard compared to our best models. The GCN-VN result is only intermediary (tuned for a smaller number of runs), but shown to confirm the positive trend.

| | Hits@100 |
|---|---|
| SAGE +attr.* | n/a |
| SAGE +dist.* | n/a |
| Adamic Adar* | $0.3245 \pm 0.0000$ |
| SEAL* | $0.4880 \pm 0.0316$ |
| GCN* | $0.1867 \pm 0.0132$ |
| SAGE* | $0.1655 \pm 0.0240$ |
| GIN | $0.2749 \pm 0.0082$ |
| GCN-VN | $0.1991 \pm 0.0062$ |
| GCN-CM$^+$ | $0.2388 \pm 0.0047$ |
| GIN-CM$^+$ | $0.2556 \pm 0.0079$ |

We used 500 epochs with a patience of 30. Furthermore, for `collab`, we used the validation edges during testing (OGB contains both settings, with and without them).

# E ADDITIONAL EXPERIMENTAL RESULTS

## E.1 RESULTS ON `ppa`

The `ppa` dataset is challenging in both its size and density. Since we were missing the resources to run experiments for all baselines on this dataset, we compare our best models (trained only on `ppa10`, we did not do additional hyperparameter tuning) to the OGB leaderboard in Table 7. For GCN, we see that our virtual node approach is able to improve the results considerably – even if only trained on 10% on the data.

Table 8: Additional results on small datasets.

| | cora
Hits@10
10 runs | pubmed
Hits@20
10 runs |
|---|---|---|
| GCN | $0.9507 \pm 0.0309$ | $0.9675 \pm 0.0143$ |
| - VN | $0.9211 \pm 0.0324$ | $0.9579 \pm 0.0214$ |
| - RM | $0.8818 \pm 0.0472$ | $0.9522 \pm 0.0110$ |
| - RM-f | $0.8858 \pm 0.0653$ | $0.8100 \pm 0.0781$ |
| - CM | $0.9100 \pm 0.0477$ | $0.9575 \pm 0.0230$ |
| - CM$^+$ | $0.8239 \pm 0.1254$ | $0.9189 \pm 0.0514$ |
| SAGE | $0.8911 \pm 0.0768$ | $0.9779 \pm 0.0105$ |
| - VN | $0.9167 \pm 0.0689$ | $0.9659 \pm 0.0333$ |
| - RM | $0.8672 \pm 0.0446$ | $0.9433 \pm 0.0208$ |
| - RM-f | $0.9116 \pm 0.0771$ | $0.9800 \pm 0.0087$ |
| - CM | $0.9234 \pm 0.0487$ | $0.9834 \pm 0.0068$ |
| - CM$^+$ | $0.8057 \pm 0.0609$ | $0.9754 \pm 0.0139$ |
| GIN | $0.8473 \pm 0.0828$ | $0.9234 \pm 0.0166$ |
| - VN | $0.8939 \pm 0.0344$ | $0.9790 \pm 0.0070$ |
| - RM | $0.7744 \pm 0.0799$ | $0.9604 \pm 0.0158$ |
| - RM-f | $0.8505 \pm 0.0836$ | $0.7986 \pm 0.0993$ |
| - CM | $0.8748 \pm 0.0380$ | $0.9125 \pm 0.0378$ |
| - CM$^+$ | $0.8419 \pm 0.0600$ | $0.9037 \pm 0.0262$ |

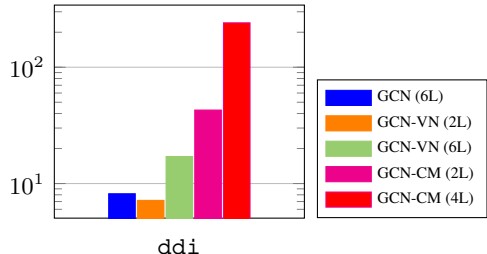

Figure 5: Performance depending on layers: time per epoch (sec.); `ddi`. We observe from Figure 2 that we get better Hits@20 for the virtual nodes models already at 2 layers than for GCN at 6 layers (its best score), hence we also compare these models. Here we see that a single virtual node can have a positive impact at the same time on both prediction scores and efficiency. The clustering takes more time.

### E.2 RESULTS ON `cora`

We ran the models also on the small `cora` data, yet the results confirm our expectation, that virtual nodes for link prediction should be used in challenging graphs. In contrast, for `cora`, we get already good scores with a regular GCN. See Table 8.

### E.3 RUNTIME

We show the runtimes on `ddi` in Figure 5. Here we see that a single virtual node can have a positive impact at the same time on both prediction scores and efficiency, while the clustering takes more time.

### E.4 COMPARISON OF CLUSTERING ALGORITHMS

See Table 9 and analysis in the main paper.

Table 9: Comparison of clustering algorithms to determine virtual node connections; Hits@20, `ddi`. See analysis in the main paper.

|  | GCN | SAGE | GIN |
|---|---|---|---|
| $\text{CM}_{metis}$ | $0.6322 \pm 0.1565$ | $\mathbf{0.8819 \pm 0.0341}$ | $\underline{0.6544 \pm 0.0960}$ |
| $\text{CM}^{+}_{metis}$ | $\underline{0.6554 \pm 0.1543}$ | $0.8041 \pm 0.1058$ | $0.5962 \pm 0.1575$ |
| $\text{CM}_{graclus}$ | $0.6324 \pm 0.1048$ | $0.5406 \pm 0.3769$ | $\mathbf{0.7299 \pm 0.0604}$ |
| $\text{CM}_{diffpool}$ | $\mathbf{0.7392 \pm 0.0381}$ | $0.7556 \pm 0.1252$ | $0.5436 \pm 0.1572$ |

Table 10: Result averaged over 10 runs, 3 run averages for comparison. We see that the results are relatively stable. Due to the randomness of the random models, we did not spend the resources to compute the results for those.

|  | collab Hits@50 | |
|---|---|---|
|  | 10 runs | 3 runs |
| SAGE +attr.* | n/a | n/a |
| SAGE +dist.* | n/a | n/a |
| Adamic Adar* | $\mathbf{0.6417 \pm 0.0000}$ | n/a |
| SEAL* | $\underline{0.6364 \pm 0.0071}$ | n/a |
| DeeperGCN* | $0.6187 \pm 0.0045$ | n/a |
| GCN+JKNet* | n/a | n/a |
| GCN* | $0.4714 \pm 0.0145$ | n/a |
| GCN | $0.4921 \pm 0.0102$ | $0.4955 \pm 0.0064$ |
| - VN | $0.5077 \pm 0.0079$ | $0.5049 \pm 0.0088$ |
| - RM | n/a | $0.5083 \pm 0.0109$ |
| - RM-f | n/a | $0.5046 \pm 0.0049$ |
| - CM | $0.5195 \pm 0.0050$ | $0.5181 \pm 0.0076$ |
| - CM$^{+}$ | $0.5163 \pm 0.0063$ | $0.5128 \pm 0.0129$ |
| SAGE* | $0.5463 \pm 0.0112$ | n/a |
| SAGE | $0.5516 \pm 0.0171$ | $0.5662 \pm 0.0149$ |
| - VN | $0.5901 \pm 0.0063$ | $0.5875 \pm 0.0091$ |
| - RM | n/a | $0.5830 \pm 0.0087$ |
| - RM-f | n/a | $\mathbf{0.6067 \pm 0.0063}$ |
| - CM | $0.6017 \pm 0.0137$ | $\underline{0.6056 \pm 0.0105}$ |
| - CM$^{+}$ | $0.5965 \pm 0.0099$ | $0.5940 \pm 0.0262$ |
| GIN | $0.5696 \pm 0.0133$ | $0.5768 \pm 0.0179$ |
| - VN | $0.5736 \pm 0.0164$ | $0.5863 \pm 0.0254$ |
| - RM | n/a | $0.5412 \pm 0.0174$ |
| - RM-f | n/a | $0.5335 \pm 0.0087$ |
| - CM | $0.5742 \pm 0.0124$ | $0.5821 \pm 0.0081$ |
| - CM$^{+}$ | $0.5371 \pm 0.0207$ | $0.5557 \pm 0.0026$ |

### E.5 Additional Runs for `collab`

Table 10 compares several 10-run averages over `collab` to the 3-run averages. The numbers are stable.

## F Cluster Analysis

We computed additional statistics about our "virtual node clusters" (i.e., a cluster represents a set of nodes connected to the same virtual node). Our hypotheses was that our proposed clustering based on

Table 11: Percentage of intra-cluster test edges, using numbers of virtual nodes between 8 and 64. For RM and $CM^+$, we average over 10 different seeds but drop standard deviation (which is negligible) for readability. The numbers in brackets with -$CM^+$ are the numbers of original METIS clusters which are then merged into "virtual node clusters"; we see no great sensitivity for them. We highlight our proposed graph-based clustering models we used in the experiments.

| | ddi | | | | |
|---|---|---|---|---|---|
| | 4 | 8 | 16 | 32 | 64 |
| CM | 49.37 | 30.33 | 17.43 | 03.81 | 04.08 |
| RM / $RM^F$ | 25.03 | 12.50 | 06.26 | 03.11 | 01.57 |
| $CM^+$ (200) | 25.31 | 12.95 | 06.82 | 03.61 | 02.13 |
| $RM^F$ | 94.36 | 73.75 | 47.48 | 27.16 | 14.60 |
| $CM^+$ (200) | 94.65 | 73.51 | 47.85 | 27.06 | 14.89 |
| $CM_{graclus}$ | 100.0 | 100.0 | 100.0 | 99.95 | 99.73 |

| | collab | | | | |
|---|---|---|---|---|---|
| | 4 | 8 | 16 | 32 | 64 |
| CM | 81.01 | 75.85 | 68.29 | 59.92 | 52.10 |
| RM / $RM^F$ | 25.05 | 12.50 | 06.22 | 03.14 | 01.52 |
| $CM^+$ (200) | 58.97 | 52.38 | 48.90 | 47.12 | 46.31 |
| $CM^+$ (1000) | 53.33 | 45.63 | 41.62 | 39.69 | 38.68 |
| $RM^F$ | 94.35 | 73.76 | 47.51 | 27.21 | 14.25 |
| $CM^+$ (100) | 95.50 | 86.10 | 73.14 | 61.84 | 56.39 |
| $CM^+$ (200) | 96.68 | 85.83 | 71.55 | 59.88 | 53.01 |
| $CM^+$ (1000) | 96.42 | 82.39 | 67.36 | 54.70 | 47.09 |

| | ppa10 | | | | |
|---|---|---|---|---|---|
| | 4 | 8 | 16 | 32 | 64 |
| CM | 79.95 | 73.87 | 70.37 | 64.97 | 58.97 |
| RM / $RM^F$ | 25.01 | 12.51 | 06.25 | 03.12 | 01.57 |
| $CM^+$ (1000) | 51.58 | 43.55 | 39.24 | 37.43 | 36.34 |
| $RM^F$ | 94.39 | 73.69 | 47.51 | 27.18 | 14.61 |
| $CM^+$ (200) | 96.94 | 86.19 | 72.78 | 62.59 | 56.12 |
| $CM^+$ (1000) | 96.36 | 83.07 | 66.03 | 53.56 | 45.06 |

the graph structure better reflects the distribution of test edges than, for example, random clustering. We report the results in Table 11. For the -$RM^F$ and -$CM^+$ models we report two numbers. The upper one shows the average number of intra-cluster test edges over 10 runs. The numbers in the lower part distinguish the actual edges and reflect how many different test edges occur in a cluster over the 10 runs. These numbers hence represent lower and upper bounds respectively.

As expected, the numbers for -CM are in between those bounds. For `ddi`, we see that the -$CM^+$ and -$RM^F$ numbers are very similar, while the ones for -$CM^+$ are much better over `collab` and `ppa10`.

## G INVESTIGATION OF NODE EMBEDDINGS

We also investigated the embeddings of the virtual nodes and compared them to the ones of the regular grapoh nodes, but we could not derive many conclusions. The main finding is that the virtual node embeddings are much more diverse than the mean of the embeddings in corresponding cluster – we would have expected them to be similar.

## H    DETAILS ABOUT P-GNN

The model closest to our approach is the position-aware graph neural network (P-GNN) (You et al., 2019). It assigns nodes to random subsets of nodes called "anchor-sets", and then learns a non-linear aggregation scheme that combines node feature information from each anchor-set and weighs it by the distance between the node and the anchor-set. That is, it creates a message for each node for every anchor-set, instead of for each direct neighbor.

We ran experiments with P-GNN but did not obtain conclusive results. It did not run on the larger datasets. For ddi, we considered the number of anchor nodes as hyperparameter since the fixed choice of 64 from the experiments of (You et al., 2019) did not yield good results. However, larger numbers such as 128 or 512 resulted in very large runtimes (9 hrs / epoch). The result in Table 3 is an intermediate best value after 50 runs. We contacted the authors and they indeed mentioned that the model is not very scalable and suggested to use just the anchor-set distance as additional features, instead of overtaking the adapted message passing as well. We did not do this extra experiment since the SAGE +dist model, whose numbers we report, follows a similar approach.

