# OpenReview forum: "Revisiting Virtual Nodes in Graph Neural Networks for Link Prediction"
_ICLR.cc/2022/Conference — ICLR 2022 Submitted_

### Official Review · Reviewer_mmjp · 2021-10-31

**Correctness:** 4
**Technical Novelty And Significance:** 2
**Empirical Novelty And Significance:** 2
**Recommendation:** 6
**Confidence:** 4

**Main Review:**

Overall, the paper is well written and the proposed methods are clearly explained and are easy to understand. And the experiments covers several different perspectives of the influence of the virtual nodes. Everything is clearly introduced and I do not have any question regarding the proposed model.

However, the contribution of the paper is mainly from the relatively narrow engineering perspective (a technique to improve the performance) without much conceptual or theoretical insights. It analyze the very specific problem of adding virtual nodes to enhance the link prediction performance. Besides, the designs of different virtual nodes are a little bit ad-hoc. Therefore, the studies on problem of whether virtual nodes will help link prediction seem not systematic enough.


----------post rebuttal----------

After rebuttal, the authors explained the rationale behind using virtual nodes for link prediction to some extend. Although there is still large sapce for improvement on the analysis proposed in the paper, the attempt on analyzing virtual nodes' influence on link prediction may catch some attention and interests from the community. Therefore, I think the paper can be rated as 'marginally above the threshold'.

**Summary Of The Paper:**

This paper proposes to enhance the link prediction performance of GNNs by adding multiple virtual nodes. The vortual nodes are designed in different ways, and the experiments analyze the influence of adding virtual nodes from different perspectives.



**Summary Of The Review:**

Contribution is limited, as explained in the main review, and I think it is not enough to be accepted.

---

> ### Author Response · Authors · 2021-11-15
> **We thank the reviewer for their considerate comments! Please see our rebuttal below.**
>
>
> - **"Contribution For Very Specific Problem Lacking Conceptual or Theoretical Insights"** \
> Generally, we want to emphasize that the link prediction problem itself is important and, given its nature in the graph context, that it is particularly of interest for the GNN community. In the GNN community, virtual nodes are popular as a small trick to improve graph classification performance. The intuition is either to directly model the entire graph embedding in a single node and/or to bridge long ranges, such that messages can be passed further. However, none of the past works provides any theoretical justification. There are commonalities in the advantages of using virtual nodes for graph classification and link prediction, but the role of virtual nodes in link prediction is to improve the link instead of the node/graph representation. Hence, our theoretical analysis analyzes the influence of nodes and the impact on the structural link representation from different types of virtual nodes. To the best of our knowledge, our paper provides the first conceptual or theoretical insights for the concept of virtual nodes, and they are tailored to the link prediction scenario. It is true that we also provide extensive experiments, however, we provide those primarily to support the theory.
>
>
> - **Design of Virtual Node settings** \
> In order to compare our proposed clustering-based virtual node setting systematically to alternatives, we chose the rather intuitive existing setting using a single virtual node and different kinds of random assignments. Please note that randomized alternatives usually provide a good baseline, and also the related architectures which point out anchor nodes to compute anchor distances use random assignments. We hope that this clarifies the system behind our study.

---

> > ### Comment · Reviewer_mmjp · 2021-11-18
> > **Response to the authors' rebuttal**
> >
> > Thanks for the further clarification provided by the authors.
> >
> > I agree that the link prediction is a very important task. However, I still think that the proposed model is of limited contribution. Originally, I though the paper had enough contribution from the perspectives of engineering and experimental performance. After checking other reviews, however, I found that more than one reviewer pointed out that the performance was not better than the state-of-the-art. I also read the authors' responeses to this. Although there is possible misunderstanding regarding the name of the baselines (and I highly appreciate that the authors also show the inferior results on Cora and Pubmed), it is unquestionable that this paper is not aimed at beating the state-of-the-art methods and make great contribution experimentally.
> >
> > Therefore, I went back to check the theoretical part again, since the authors strengthened the theoretical contribution. However, it seems that the only theoretical result is that adding virtual nodes will increases the influence of a node to distant nodes. It is still not clear why would the increased influence score between distant nodes would help link prediction. Intuitively it may even hurt the performance since the embedding of each node will be affected by the distant non-similar nodes.
> >
> > Overall, I choose to keep my original rating.

---

> > > ### Author Response · Authors · 2021-11-20
> > > **Clarification of Contribution**
> > >
> > > Thank you for the timely reply!
> > >
> > > - **About SOTA** \
> > > Please observe that there is no single model which can achieve SOTA on all datasets on the OGB leaderboard. Moreover, the objective of this paper is not to achieve SOTA performance, but to advance our understanding about the value of virtual nodes in link prediction.
> > >
> > >
> > > - **Contribution** \
> > > Our goal is to prove that adding virtual nodes is a general and useful trick for many different popular GNN models. We propose techniques and show how to properly add virtual nodes to make the models more powerful in practice (i.e., the engineering contribution as you said); we observe rather stable performance increases across diverse datasets. At the same time, we also give theoretical insights in order to advance our understanding about virtual nodes. Our theory covers two aspects. First, we show that both a greater node degree and number of virtual nodes increase their influence. Since virtual nodes provide shortcuts in the graphs, *multiple* virtual nodes hence allow for capturing long-range dependencies better. Second, we show that adding multiple virtual nodes can increase the expressiveness of the structural link representation, as we described in the last paragraph of Section 4.3 and Appendix B.
> > >
> > >
> > > We hope this clarifies any misunderstanding. Please let us know if not.

---

> > > > ### Comment · Reviewer_mmjp · 2021-11-21
> > > > **On the theoretical analysis and the role of virtual nodes**
> > > >
> > > > Thanks for the further clarification from the authors.
> > > >
> > > > From the responses, I further confirmed my understanding that this paper aims to provide conceptual/theoretical understanding on how virtual nodes affect the link prediction task. Therefore, the evaluation of the paper should be mainly based on the theoretical analysis.
> > > >
> > > > For the first theoretical aspect mentioned by the authors, as I mentioned in my last response, I understand that the virtual nodes create shortcuts between distant nodes and allows for capturing long-range dependencies. But why would this help the link prediction task? It makes sense that the capturing of long-range dependencies is beneficial for generating graph level representations. But the link prediction seems to require local patterns, e.g. whether two nodes are similar, and integrating information from distant nodes seems not helpful To summarize, the logic from 'capturing long-range dependencies' to 'better link performance' is not straightforward, which diminishes the theoretical contribution of the paper.
> > > >
> > > > For the second aspect, I went back to check the appendix for the analysis on the expressiveness of the structural link representation. However, the analysis is actually on expressiveness of node representations. I understand that the link prediction is based on the node representations and therefore will benefit from this. But if the analysis on link prediction equals the analysis on node representations, why is it necessary to specially study the influence of virtual nodes on link prediction? Actually both theoretical aspects mentioned by the authors are analysis on nodes. The first is on analyzing the influence score of a node on distant nodes, and the second is on the expressiveness of node representations. Overall, this makes the contribution of this paper ambiguous.
> > > >
> > > > To summarize, my concerns on the proposed analysis of virtual node influence on link prediction are twofold:
> > > > 1. The provided analysis on node-wise influence score change does not explain why would the link prediction task benefit from it.
> > > > 2. The provided analysis all focus on nodes, therefore it is unclear why is it necessary to specially study the influence of virtual nodes on link prediction.

---

> > > > > ### Author Response · Authors · 2021-11-23
> > > > > **Further Clarification**
> > > > >
> > > > > We try to further explain, we can definitely update explanations in the paper as well.
> > > > >
> > > > > - **(Q1) "The logic from 'capturing long-range dependencies' to 'better link performance' is not straightforward"**
> > > > > \
> > > > > \
> > > > > First, observe that the growing impact of a virtual node on the connected nodes is here indeed first of all in a "local" form in our case, since the METIS clustering groups local neighborhoods.
> > > > > \
> > > > > \
> > > > > Nevertheless, there is quite some evidence that capturing long-range dependencies increases link prediction performance of GNNs.
> > > > > First, it is well known that the successful link prediction heuristics are mostly dependent on the paths between the target nodes of the link, and these paths are not necessarily short [1].
> > > > > The importance of these paths can also be illustrated by a hypothetical scenario without node features: if there is no information about the connection(s) between them, link prediction boils down to random guessing.
> > > > > Second, graph neural networks that achieve SOTA performance in link prediction often exploit concepts such as diffusion [2] or large numbers of layers [3], which specifically allow for capturing longer ranges.
> > > > > \
> > > > > \
> > > > > [1] Zhang et al. Link Prediction Based on Graph Neural Networks, NeurIPS 2018.\
> > > > > [2] Wang et al. Multi-hop Attention Graph Neural Networks, IJCAI 2021.\
> > > > > [3] DeeperGCN: All You Need to Train Deeper GCNs, arxiv 2020.
> > > > >
> > > > >
> > > > >
> > > > > - **(Q2) Choice of Link Prediction Task**
> > > > > \
> > > > > Given the practical relevance and theoretical challenges of the link prediction problem, it has been the focus in many works about GNNs recently [4,5,6].
> > > > > \
> > > > > \
> > > > > Nevertheless, we never intended to limit our theory to link prediction. In fact, we are confident that our results also apply to node classification. However, to show this, we need to adapt the theories (e.g., the analysis and numbers in Step 2 in the proof of Theorem 1 will change), and further experiments which are out of the scope of this paper.
> > > > > \
> > > > > \
> > > > > In line with the majority of AI papers, our goal is not only to provide conceptual/theoretical understanding, but to justify a (an adaptation of a) new practical model or technique by providing a theoretical investigation. Experiments cannot prove but only confirm theory, and ours were designed such that the results likely generalize to different kinds of data, and therefore also focus on a consistent task. We note that this is still not usual, our evaluation shows that many of the recently proposed rather complex models are hardly applicable on the more realistic data. Interestingly, we can see our theoretical hypotheses generally confirmed on this data: on challenging, larger graphs, graph density & multiple virtual nodes increase GNN link prediction performance for a number of popular GNNs.
> > > > > \
> > > > > \
> > > > > Overall, we think our experimental contribution provides not only enough, but really good evidence for the very first study about virtual nodes in larger graphs, whose goal is to motivate both practical usage and further investigation (e.g., node classification, other architectures).
> > > > > \
> > > > > \
> > > > > [4] Cai et al. A Multi-Scale Approach for Graph Link Prediction, AAAI 2020. \
> > > > > [5] Li et al. Distance-Enhanced Graph Neural Network for Link Prediction, ICML 2021.\
> > > > > [6] Zhang et al. Labeling Trick: A Theory of Using Graph Neural Networks for Multi-Node Representation Learning, NeurIPS 2021.

---

> > > > > > ### Comment · Reviewer_mmjp · 2021-11-30
> > > > > > **Further response**
> > > > > >
> > > > > > Sorry for the late response.
> > > > > >
> > > > > > The further responses provided by the authors to some extent explained why 'long range dependency' would benefit link prediction, but the explaination seem not suitable for virtual nodes. As explained by the authors, the link prediction may highly rely on the paths between two nodes. This is perfectly reasonable. However, the long range denpendency enabled by virtual node does not include the path information between two nodes, instead, the information between two nodes are directly exchanged via the virtual nodes regardless of the path between the two nodes. Therefore, it is still clear how the long range dependency enabled by virtual nodes help the link prediction.
> > > > > >
> > > > > > As for the second response. I totally agree that the link prediction task is important and I'm not asking whether the proposed framework can be applied to other tasks such as node related tasks. What I am concerned is that the analysis for link prediction are all focused on nodes. Therefore, it seems that the analysis of link prediction is just same as the analysis for nodes, which weakens the contribution of this paper.
> > > > > >
> > > > > > Finally, my recommendations on revising this paper is to make the analysis focused on links instead of on nodes. Even if the virtual nodes first influence the node representation and therby benefit the link prediction, the analysis should not stop at analyzing the influence on node representation learning. A further step towards link prediction is necessary. Besides, the current explanation based on long range dependency is not convincing, as mentioned above.

---

> > > > > > > ### Author Response · Authors · 2021-11-30
> > > > > > > **On the Contribution**
> > > > > > >
> > > > > > > Thank you for getting back to us! We comment on the first part in our main reply. \
> > > > > > > Regarding the second part, please consider the work in context of the field:
> > > > > > >
> > > > > > > - **"Analysis for nodes weakens the contribution of this paper"**
> > > > > > > \
> > > > > > > We do not fully understand why a completely new analysis which does not exist in related work in these aspects (i.e., there is neither such analysis on nodes or in the context of node classification) weakens the contribution of the paper just because it focuses on nodes. We mentioned above that the link representation based on nodes is standard in the field and used in basically any paper that uses a GNN for link prediction (e.g., in the original paper paper on GCNs by Kipf et al. but also in the most recent ones). The recommendation seems to imply that we should stop all GNN research/publishing on link prediction until someone comes up with a more interesting link representation which is not primarily based on nodes. While we are not able to come up with such a representation directly, we chose to bring more understanding in the field by analyzing the virtual node technique in terms of the influence score. We agree that the long-term goal (for the entire field) is a link representation that goes beyond the concatenation of node embeddings.

---

> > > > > > > > ### Comment · Reviewer_mmjp · 2021-12-01
> > > > > > > > **Further response to the contribution**
> > > > > > > >
> > > > > > > > I'm concerned with the contribution because the analysis in the paper is not new. The influence score analysis on the nodes was already used in multiple works (the authors also cited related works on this). And the proposed analysis was just using this existing framework to compute the influence score between nodes in a graph with several extra virtual nodes. The computation seems very straightforward without much insights. That is why I was saying that analyzing nodes is not enough and why I felt the contribution was not enough.
> > > > > > > >
> > > > > > > > I agree with the importance of link prediction task, and I also agree that the attempt of this paper to analyze the influence of virtual nodes on link prediction is valuable. However, in my opinion, the analysis in this paper is mostly repeating existing analysis on nodes. That is why I kept talking about further adapting the analysis to link prediction. I don't mind the analysis is based on nodes or the analysis is based on existing works, but I do think there should be something new about links.
> > > > > > > >
> > > > > > > > I understand that the evaluation on whether the contribution is enough may be subjective, but unfortunately I have to give judgements based on my limited experiences.

---

> > > > > > > > > ### Author Response · Authors · 2021-12-01
> > > > > > > > > **On the Novelty of the Theoretical Analysis**
> > > > > > > > >
> > > > > > > > > Thank you for getting back to us!
> > > > > > > > >
> > > > > > > > > The goal of the paper is to show theoretically and empirically that (multiple) virtual nodes are a (new,) simple and effective technique which can be used for link prediction - and most likely also node classification, this is to be proven experimentally.
> > > > > > > > >
> > > > > > > > > For this we apply (actually two) proven, existing theoretical frameworks, the influence score and the labeling trick, in order to analyze our proposal from different perspectives.
> > > > > > > > > - These frameworks were developed for reuse (i.e., peer-reviewed, proven, etc.), starting at zero each time is not very productive. As the reviewer notes, they have been applied in more than one work already.
> > > > > > > > > - Finding the right techniques to apply and applying them is not trivial either (e.g., the proof in Step 2 in Appendix B is completely new). You cannot conclude that a work reusing existing mathematics does not provide new insights.
> > > > > > > > > - Finally, the interpretation in terms of recognized existing techniques is interesting in itself, and connects it to other works. In fact, Reviewer gjKs mentions it under strengths (see also Reviewer Guidelines 5.5).
> > > > > > > > >
> > > > > > > > > In summary, we re-emphasize that the innovation of our paper lies in increasing our theoretical understanding of the virtual nodes technique and motivating its application (in an adapted version) for other problems. Amongst others, we conducted an extensive empirical evaluation. We see the reviewer's point, but we do not think re-inventing the wheel should be necessary to get a paper accepted, other researchers may be interested in the results though. Maybe you could re-consider this point. We also want to stress it for the overall discussion, and hope others decide differently.

---

> > > > > > > > > > ### Comment · Reviewer_mmjp · 2021-12-02
> > > > > > > > > > **Further response**
> > > > > > > > > >
> > > > > > > > > > Thanks for the further explanations. I can feel the anxiety of the authors and I understand my careful consideration on the paper matters a lot to the authors. So I did reconsidered the contribution of the paper. I think it is acceptable to increase the rating from 5 to 6, since the influence of virtual nodes on link prediction is under-explored and early attempts on this may not be polished enough.
> > > > > > > > > >
> > > > > > > > > > But the authors may have misunderstood my points. In my last comment, I said 'I don't mind the analysis is based on nodes or the analysis is based on existing works, but I do think there should be something new about links.'. Using existing works is totally acceptable, and the lack of insights is not caused by reusing existing techniques. I have clearly explained what is lacked in the previous comments, and the authors may look back if are interested.
> > > > > > > > > >
> > > > > > > > > > Good luck!

---

> > > > > > > > > > > ### Author Response · Authors · 2021-12-02
> > > > > > > > > > > **Thank you!**
> > > > > > > > > > >
> > > > > > > > > > > We sincerely thank you for taking the time for this extended and fair discussion.
> > > > > > > > > > >
> > > > > > > > > > > We totally agree on the point that link representations beyond the ones we consider are one important research goal the community should focus on, and we will investigate in this direction as well.

---

### Official Review · Reviewer_CjZq · 2021-11-02

**Correctness:** 3
**Technical Novelty And Significance:** 2
**Empirical Novelty And Significance:** 3
**Recommendation:** 6
**Confidence:** 4

**Main Review:**

Strength:
1.	By the authors’ claim, this is the first work to employ virtual nodes to improve the link prediction tasks.
2.	Virtual nodes are well motivated to capture long-distance / under-reaching messages between nodes. The authors provide both theoretical and empirical analysis for the virtual node setting.
Weakness/concerns:
1.	The theoretical analysis is limited to regular graph for influence score and non-attribute graph for expressiveness of link representation. Could them be generalized to more applicable graphs?
2.	The authors concatenate node representation as link representation. In this way, the expressiveness of link representation is highly related to the expressiveness of node representation. Therefore, it seems powerful GNNs for node representation or node classification can be directly used for link representation or link prediction. But it seems that P-GNN conflicts with this claim as the performance of P-GNN is really bad though the authors mention some concerns about it in supplements.
3.	As the experimental results show, virtual nodes do not always benefit the link prediction, such as on Cora and Pubmed. Although the authors give some analysis, readers may still be confused about in what situations virtual nodes are recommended and vice versa. I would appreciate if the authors can give a table to further explain on it, especially clarify ambiguous expression in the article. For example, what “cora/pubmed have no cluster structure” means, when both Cora and Pubmed have clearly defined classes and previous works have shown that their data points have underlying clusters.
4.	The proposed method seems to rely heavily on sophisticated hyperparameter searching. (as shown in Appendix D)


**Summary Of The Paper:**

The authors revisited the commonly used trick of virtual nodes in graph learning. The authors proposed the multiple virtual nodes usage under the link prediction scenario and provided both theoretical and empirical supports for it. For theoretical analysis, the authors consider the influence score for m-regular graph and expressiveness of link representation (by concatenating representation of two nodes) in a special case and non-attributed graphs. For empirical analysis, the authors compare the performance of multiple virtual nodes setting to only one node setting with different GNN strategies and different datasets. They finally conclude that the virtual nodes can stably improve base GNN performance on some challenging link prediction tasks.

**Summary Of The Review:**

The authors present much interesting and insightful analysis for virtual nodes on both theoretical and empirical sides. But there are also many concerns on both two sides which should be well addressed.

---

> ### Author Response · Authors · 2021-11-15
> **We thank the reviewer for their interesting comments! Please see our rebuttal below.**
>
>
> - **(Q3) Generalizability towards Other Graph Types** \
> As it is usual (see, e.g., [#1]), our theoretic analysis focuses on the most general (and simple) setting, but the conclusion on more general graphs should be similar. Regarding a more specific graph type in Section 4.3 of our paper only results in other numbers in Equations (5),(6) and (7), their structure remains the same. Most importantly, when all node degrees are large enough, the conclusion from Equation (6) will not change at all: in a graph that is dense overall, the virtual nodes will always increase the influence score. In the analysis for structural link representation (Appendix B), we use non-attributed graphs only as an evidence (or an extreme case) that a typical 1-WL-GNN is not expressive enough and using virtual nodes can make it more powerful. \
> [#1] Vashishth et al. Composition-based Multi-Relational Graph Convolutional Networks, ICLR 2020.
>
>
> - **(Q4) Applicability of Node Classifiers for Link Prediction** \
> It is true that node classifiers are generally usable in such a link prediction approach (i.e., where we concatenate node embeddings). P-GNN was actually used for link prediction similarly, the issues we had were due to the fact that its message passing is highly complex and does not scale well to larger graphs.
>
>
> - **(Q5) Clarification about Results on Cora/Pubmed** \
> Our goal of using virtual nodes for link prediction is to enhance information propagation in large, complex graphs (see also "Clarification of Motivation" in the reply to reviewer gjKs). We conducted the experiments on the smaller Cora and Pubmed datasets primarily out of curiosity, to see how the models perform in smaller settings; and, for Pubmed, we see that we sometimes can still obtain improvements. We are aware that the results on the Cora dataset seem to contradict our proposal, but we did not want to withhold them from the community. In fact, we added them to the paper (appendix) to prevent users from expecting benefits in the wrong settings. On Cora, the standard GCN achieves already an extremely high score of 95% (i.e., compared to its performance in more challenging settings), hence it is not surprising that it does not benefit from augmentation approaches. In the updated version, we clarify this, and also removed the wrong statement about the cluster structure. Thank you for pointing this out!
>
>
> - **(Q6) Hyperparameters** \
> We considered a wide range of hyperparameters because we originally had no experience with these OGB datasets and because the original OGB paper only considers extremely few parameters (e.g., a fixed number of 2 or 3 GNN layers). Regarding the latter, we wanted to investigate if there is potential for the standard GNNs to obtain better scores, even without virtual nodes, because we suspected that the results on the leaderboards are not fair - if compared to the SOTA - in the sense that these GNNs actually can do better. For example, observe that our (Graph)SAGE obtains a Hits@20 of 0.6173 ± 0.1068 on DDI, while the leaderboard states 0.5390 ± 0.0474 (the score from the OGB paper which originally was intended to be used in a basic comparison instead of in the comparison to the SOTA). \
> It is not the case that all these hyperparameters are critical. During tuning, it turned out that we could have fixed the hidden dimension to 256, learning rate to 0.0001, and dropout to be mostly 0.3 (only on the small graphs 0 was better). We added a note to Appendix D now, and it will be clear from our GitHub repository. Hence, we are only left with the number of layers as well as virtual nodes. The former is natural to GNNs and related to the data type studied, and the latter was of interest given that we study the virtual nodes in the paper. \
> Apart from these parameters, Appendix D also lists the parameters for specific baselines, but those are not used in our GNN extensions.

---

> > ### Author Response · Authors · 2021-11-24
> > **Discussion period ends soon**
> >
> > Dear Reviewer CjZq:
> >
> > The discussion period will end soon but we have not received your feedback. We hope our reply removed the remaining concerns you had. Please let us know in case you need further clarifications.
> >
> > Thank you!

---

### Official Review · Reviewer_gjKs · 2021-11-02

**Correctness:** 4
**Technical Novelty And Significance:** 3
**Empirical Novelty And Significance:** 2
**Recommendation:** 5
**Confidence:** 4

**Main Review:**

Strengths
- Unlike most of the proposed methods for GNN focusing on node classification or graph-level tasks, the idea of ​​applying the concept of virtual nodes to link prediction is novel and interesting.
- The authors show the relationship between the proposed method and the distance encoding and labeling trick, which are popular techniques for link prediction.
- The authors theoretically analyze the effectiveness of virtual nodes in terms of influence scores.

Weakness of the paper:
- The motivation of using virtual nodes for link prediction is not clear. The authors only argue that the fact that virtual nodes have not been studied in link prediction is because the large and heterogeneous graphs in link prediction are of very different nature, but do not clearly explain why virtual nodes are important for link prediction.
- Even though the authors provide first guidelines about using virtual nodes for link prediction, experimental results are marginal or even worse compared to the results of GNNs w/o virtual nodes. Even in the ddi dataset, which showed great improvement, there is little difference from the model using distance encoding. The authors should show a more significant performance improvement or clearly show problems that using virtual nodes only addresses.


**Summary Of The Paper:**

This paper investigates using virtual nodes in graph neural networks for link prediction. Specifically, the authors use a graph clustering algorithm to determine groups of nodes in the graph and adopt multiple virtual nodes in the graph for the link prediction senario. They also theoretically investigate the effect of using virtual nodes for link prediction. Experiments conducted on six datasets  provide insights and guidelines about using virtual nodes for link prediction.



**Summary Of The Review:**

I feel that the paper gives an interesting direction to consider virtual nodes for link prediction, but the motivation is unclear and experimental results are insufficient.

---

> ### Author Response · Authors · 2021-11-15
> **We thank the reviewer for their detailed comments! Please see our rebuttal below.**
>
>
> - **Clarification of Motivation** \
> Our paper is motivated by two ideas complementing each other.
> On the one hand, and as outlined in the introduction, under-reaching represents a considerable problem for standard GNNs in larger, more complex graphs since the 2-3 layers usually used cannot propagate information between nodes that are farther apart.
> On the other hand, adding a single virtual node is a popular, simple and effective technique to improve the graph classification of GNNs; amongst other, the intuition behind is to bridge long ranges, such that messages can be passed further.
> Therefore we want to provide deeper insights and ask:
> Is adding virtual nodes also beneficial for link prediction, and/or how do we have to adapt the approach to large networks, where a single virtual node is likely not enough to store all information relevant for propagation?
> While previous works only apply the virtual node trick, our goal is specifically to give a thorough theoretical justification supported by practical guidelines.
> We give additional details about the extent of our contribution in the reply to reviewer mmjp.
>
>
> - **"Insufficient Results" in Comparison to SOTA**  \
> Our work claims that using virtual nodes represents a simple but effective technique which can be used with many basic GNN models to enhance their link prediction performance.
> We provide Table 3 in order to show that the virtual node extension yields results comparable to the state of the art. We do not think that it is necessary to outperform other, much more specialized models. We now realized that the name SAGE+attr. as it appears on the leaderboard is misleading since the authors actually also adapted the convolution operation such that it has less in common with the original GraphSAGE model and hence should be considered as its own architecture. It is true that there is (a single) comparable result for GraphSAGE extended by the anchor distance (SAGE+dist.). However, it is open how this generalises to other GNNs and datasets. More importantly, the anchor distance is computed according to a fixed scheme while the virtual node embeddings in our model are learned entirely. We believe that the latter aspect clearly delimits our approach from this related work and that is interesting in itself. We now adapted the paper in order to avoid further confusion.

---

> > ### Author Response · Authors · 2021-11-24
> > **Discussion period ends soon**
> >
> > Dear Reviewer gjKs:
> >
> > We have tried our best to address your concerns and we would like to know if there are further questions. In order to address your potential future questions in time, we hope to hear from you soon.
> >
> > Thank you!

---

> > > ### Comment · Reviewer_gjKs · 2021-11-29
> > > **Response to the authors' rebuttal**
> > >
> > > I appreciate the authors' response.
> > > After reading the author's response and other reviewer's comments, It is still questionable why virtual nodes are effective for link prediction. As authors mentioned in the discussion with reviewer mmjp, the successful link prediction heuristics are mostly dependent on the paths between the target nodes of the link, and these paths are not necessarily short. However, the reason why the path-dependent heuristics are successful can be that there is a high probability that most node pairs whose links exist are relatively close. If so, although virtual nodes can be helpful for long-distance node pairs, I'm concerned that it will rather be a hindrance to most of the local node pairs with links.
> > > Overall, I would keep my rating.

---

> > > > ### Author Response · Authors · 2021-11-30
> > > > **Further Clarification**
> > > >
> > > > Thank you for getting back to us!
> > > >
> > > > - **"It is still questionable why virtual nodes are effective for link prediction. "**
> > > > \
> > > > We give more explanation on our theory in the reply in the main thread.
> > > >
> > > > - **Impact of Longer Ranges**
> > > > \
> > > > We agree that we cannot make any general statements about how critical are longer ranges ("are successful *can be* ... *If so*, ...") since this highly depends on the kind and amount of data. However, we do think that the models should have the capability to capture them, in case they are critical. In Appendix B we show that virtual nodes provide such a capability.
> > > >
> > > > - **"I'm concerned that it will rather be a hindrance to most of the local node pairs with links."**
> > > > \
> > > > Please observe that our analysis on the influence score does not assume longer ranges. The nodes are influenced most by the virtual nodes directly connected to them which capture primarily information about their close neighborhood.

---

> > > > > ### Author Response · Authors · 2021-12-03
> > > > > **Question**
> > > > >
> > > > > Dear Reviewer gjKs,
> > > > >
> > > > > Please let us know in case anything remains unclear or concerning. We have concluded the discussion with reviewer mmjp and addressed your other comments.
> > > > > Thank you!

---

### Official Review · Reviewer_RCd9 · 2021-11-02

**Correctness:** 4
**Technical Novelty And Significance:** 3
**Empirical Novelty And Significance:** 4
**Recommendation:** 6
**Confidence:** 3

**Main Review:**

Strengths:

1. I agree with the paper that virtual nodes lack a better understanding. It is good to see that extensive experiments are conducted to evaluate several virtual node strategies and GNN-CM can improve performance in many cases.
2. The paper theoretically analyses the effect of virtual nodes on influence distributions

Concerns:

I'm particularly interested in how to decide the number of virtual nodes because it is important for practical use.
In Table 6 in the appendix, GNN-CM uses a high number of virtual nodes, but in Figure 3 the trend seems different.
In Figure 3, on ddi, GNN-CM with 2 virtual nodes works the best.
Is there any practical guidance?

By the way, some sentences need to be checked. For example, at the beginning of section 4, "......a relationship that might strongly influenced depend on surrounding relations"

**Summary Of The Paper:**

This paper analyses the roles of virtual nodes in the link prediction problem.
Extensive experiments are conducted to support the claims and show that virtual nodes can improve the link prediction performance of GNN.

**Summary Of The Review:**

A good paper with theoretical analysis and extensive experiments

---

> ### Author Response · Authors · 2021-11-15
> **We thank the reviewer for their thoughtful comments! Please see our rebuttal below.**
>
> **Number of Virtual Nodes**
>
> As you mention, Table 6 in the appendix confirms that GNN-CM generally performs better using a high number of virtual nodes, and this supports our theoretical investigation. In Figure 3, this is the case for SAGE-CM with a best number of 32 overall - the best score on ddi is slightly better for 8 virtual nodes but with a much higher standard deviation. For GCN-CM, we see more variability but also best results with 64 virtual nodes. On collab. Altogether, we consider the one result for 2 layers - which is close to the second best with 64 virtual nodes as an outlier. We clarified this in the paper now. As a general guideline, we definitely suggest to start with around 32/64 virtual nodes.

---

> > ### Author Response · Authors · 2021-11-24
> > **Discussion period ends soon**
> >
> > Dear Reviewer RCd9:
> >
> > The discussion period will end soon but we have not received your feedback. In case you have further questions, please let us know!
> >
> > Thank you!

---

> > > ### Comment · Reviewer_RCd9 · 2021-11-29
> > > **I will keep my rating**
> > >
> > > I thank the authors for their response.
> > >
> > > I have also read other reviewers' comments.
> > >
> > > Overall, I will keep my rating

---

> > > > ### Author Response · Authors · 2021-11-29
> > > > **Question on Review**
> > > >
> > > > Thank you for your response!
> > > >
> > > > We would be glad if you could let us know, what is missing for getting an "accept" rating. We originally felt quite encouraged by the positive review and wonder about aspects we were missing or which need further clarification. Thank you so much for taking the time for this discussion.

---

> > > > > ### Comment · Reviewer_RCd9 · 2021-11-29
> > > > > **To be more specific**
> > > > >
> > > > > To be more specific, I think some concerns of other reviewers (e.g., results, theoretical analysis) are also reasonable.
> > > > >
> > > > > In fact, I also feel that the results do not show a clear enough trend, but I understand that trying new things often gets mixed results.

---

> > > > > > ### Author Response · Authors · 2021-11-29
> > > > > > **Additional Clarification**
> > > > > >
> > > > > > We have given our best in the replies to the concerns of the other reviewers and are unsure which part of the theory is not clear.
> > > > > >
> > > > > > - **"Results Missing Clear enough Trend"**
> > > > > > \
> > > > > > It is true that there are larger differences between the combinations with GCN, SAGE, and GIN. However, the latter models alone perform also different on various datasets, in general, so it is not too surprising for us that we cannot draw a single perfect conclusion for the various combinations. Overall, we can see the benefits of adding virtual nodes. Further note that, for example, in Table 3, one mentioned outlier-result for GCN-CM_OL is extremely close to the one of GCN-CM (in particular, considering standard deviation).

---

### Author Response · Authors · 2021-11-15
**General Response**

We thank all reviewers for the thoughtful and fair reviews! We are grateful that the reviewers mostly recognized the novelty of the topic, the clarity of writing, the theoretical contribution, and the great extent of the experiments.

We have addressed specific comments and suggestions in the individual responses and also updated the paper accordingly (also fixing the minor issues not mentioned below). In summary, we provided more background information in order to address the concerns of reviewers gjKs and mmjp, and commented on the practical considerations discussed by reviewers CjZq and RCd9. We hope that we removed the previously remaining concerns and are happy to provide further information if needed.

---

### Author Response · Authors · 2021-11-30
**Clarification of Theory**

Given that the actual discussion started extremely late (few hours before the main deadline), we hope the reviewers consider our reply and, if necessary, follow-up discussion before deciding about the score!

Since there seems some confusion about the importance of long-range dependencies: Please note that the paper only mentions them in the motivation and in the context of GNN-CM+. Generally the importance of these dependencies depends on the data. We mentioned them together with our theory in our reply to reviewer mmjp on 11/20 because we find them illustrative. However, we did not intend to cause confusion.


**The main part of our theory** focuses on the impact of node degree and number of virtual nodes on the node representation. In particular, we show that increases in the former yield increases in the influence of the virtual nodes. Hence, (1) if we choose the virtual node assignment appropriately and (2) if we can learn good virtual node embeddings (i.e., collecting relevant information of the connected nodes and paths), these embeddings can help improve prediction performance. We assume (2) and provide a solution to (1), and our experiments show that we can indeed learn useful virtual node embeddings.

Observe that the nodes are influenced most by the virtual nodes directly connected to them which capture primarily information about their close neighborhood if we assume a distance-based clustering.

**The second part of our theory** (Appendix B) actually gives some explanation about long-range dependencies from the viewpoint of link prediction, by investigating them in the context of the "labeling trick" [1]. Assuming the clustering is based on distance: if two nodes are farther apart in the graph, they are clustered differently and hence can help to recognize non-isomorphic links.

[1] Zhang et al. Labeling Trick: A Theory of Using Graph Neural Networks for Multi-Node Representation Learning, NeurIPS 2021.

---

### Decision · Program_Chairs · 2022-01-20

**Decision:**

Reject

**Comment:**

This paper considers GNNs for link-prediction (predicting which links are likely to appear next). An idea that has been used before is to add virtual nodes to improve the ``under-reaching” problem in shallow GNNs; this paper considers this systematically in the context of link prediction. Specifically, one approach developed is to cluster the graph into clusters C(i), I = 1, 2, …, k for some k and to add a virtual node u(i) for each index i, which is made adjacent to each node in C(i). This can ease information exchange, particularly in message-passing GNNs.

Link prediction is an important problem. However, there seem to be at least three issues with this work: the performance gains obtained are not strong enough, it is not conceptually clear why virtual nodes should help with link prediction, and the analysis is quite a bit about repeating existing analyses on nodes alone. I recommend the authors to address these issues thoroughly in the next version of the paper.